# A Quinoxaline−Naphthaldehyde Conjugate for Colorimetric Determination of Copper Ion

**DOI:** 10.3390/molecules27092908

**Published:** 2022-05-03

**Authors:** Sutapa Sahu, Yeasin Sikdar, Riya Bag, Michael G. B. Drew, José P. Cerón-Carrasco, Sanchita Goswami

**Affiliations:** 1Department of Chemistry, University of Calcutta, 92, A.P.C. Road, Kolkata 700009, India; sutapa.sahu11@gmail.com (S.S.); y.sikdar@gmail.com (Y.S.); riyabag.chem@gmail.com (R.B.); 2Department of Chemistry, The Bhawanipur Education Society College, 5, Lala Lajpat Rai Sarani, Kolkata 700020, India; 3Department of Chemistry, University of Reading, Whiteknights, Reading RG6 6AD, UK; m.g.b.drew@reading.ac.uk; 4Centro Universitario de la Defensa, Academia General del Aire, Universidad Politécnica de Cartagena, C/Coronel López Peña S/N, Santiago de La Ribera, 30720 Murcia, Spain

**Keywords:** quinoxaline−naphthaldehyde conjugate, Cu^2+^ colorimetric detection, absorption study, ab initio calculations

## Abstract

This work facilitates detection of bivalent copper ion by a simple Schiff base probe QNH based on a quinoxaline−naphthaldehyde framework. The detailed study in absorption spectroscopy and theoretical aspects and crystal study of the probe and probe−copper complex has been discussed. The detection limit of the probe in the presence of Cu^2+^ is 0.45 µM in HEPES−buffer/acetonitrile (3/7, *v*/*v*) medium for absorption study. The reversibility of the probe−copper complex has been investigated by EDTA. The selective visual detection of copper has been established also in gel form.

## 1. Introduction

Metals are unavoidable particulates in our surroundings and contain a wide panel of industrial applications [1,2,3]. However, metals are also related to our living systems, those of which include transition metals, alkali metals, and alkaline earth metals during biological machinery [4,5]. They are present in environment as well as biological systems in a very traceable form that is not easy to distinguish. The main concern about the metal ions is that they may be toxic if present in a higher amount than their permissible limit [6]. Detecting metal ions in very trace amounts seems to be a hurdle for researchers, so continuous experiments are in progress on this particular topic.

Cu^2+^ was one of the most found elements after Fe^3+^ and Zn^2+^ on Earth’s surface. A trace amount of Cu^2+^ is considered an essential nutrient for the human body [7]. Cu^2+^ is the main component of a variety of enzymes such as superoxide dismutase (SOD1), cytochrome−c−oxidase (Cyt−O), tyrosinase, etc [8]. Its versatile role in chemistry, biochemistry, industry, and the medicinal field allows it to easily contaminate our environment [9]. Excess consumption of Cu^2+^ in the human body can lead to serious health issues, such as Wilson’s disease [10,11], Alzheimer’s disease [12], Menkes disease [13], Parkinson’s disease [14], transmissible spongiform encephalopathy [15], etc. Due to its wide usage, it is crucial to discover new strategies for detecting trace amounts of Cu^2+^ readily and in an inexpensive way.

Colorimetric chemosensors are preferable for detection of analytes, as no expensive tools are needed for the experiments which, in turn, results in a more effective and less time–consuming detection than other methods [16,17]. This can be recognised by the naked eye and easy monitoring can be possible. Several works have been reported for colorimetrically Cu^2+^ detection (Appendix A) [9,18,19,20,21,22,23,24,25,26,27].

Quinoxaline moiety is one of the units that has recently grabbed our attention due to its excellent colorimetric signalling unit behaviour [28,29]. The two N atoms present in the ring not only extend the conjugation in the ring, but are also preferred as a good chelating unit [28]. Schiff base moiety with extended conjugation and –OH and –NH binding sites are effective for photoactive detection of metal ions [30,31].

Quinoxaline and naphthaldehyde-based Schiff base chemosensors (QNH) have been reported in literature by Das and co-workers for colorimetric detection of Cu^2+^ in methanol/aqueous HEPES−buffer (9/1, *v*/*v*) [31]. Aiming at further expanding the practical use of that new probe, we herein perform a joint experimental and theoretical study of QNH in HEPES−buffer/acetonitrile (3/7, *v*/*v*) medium for Cu^2+^. The reversibility of the QNH with Cu^2+^ in the presence of EDTA has been also assessed. The QNH + Cu^2+^ complex was further exposed to a different amino acid solution and shows a response in the presence of histidine.

## 2. Results and Discussion

### 2.1. Synthesis and Characterization

The probe QNH was synthesized in two simple steps, as discussed in the experimental section. In the first step, the heterocyclic quinoxaline moiety with ketone functionality (compound **1**) had been synthesized, and in the second step reaction with the 2–hydroxy–1 naphthaldehydehydrazide (compound **2**), the condensed product we found in orange powder-like form in 73% yielded in methanol, as illustrated in Appendix A. QNH crystallized by layering technique in CHCl_3_ and diethyl ether solvent and the dark orange diamond-shaped single crystal we obtained was further used in characterization and spectral studies.

The probe is fully characterized in terms of CHN analysis, FT−IR, ^1^H NMR, ESI−MS spectral analysis (Appendix A), and single–crystal XRD method before its application. The FT−IR spectrum confirms the peak at 1623 cm^−1^ for the generation of imine bond (C=N). The ^1^H NMR spectrum shows peaks at δ_H_ 8.2–7.3 for aromatic protons, δ_H_ 9.86 for aldehyde proton, and δ_H_14.1 for the –OH proton. The ESI−MS spectrum shows peak at 401.14 amu for the [C_26_H_17_N_4_O]^+^ species. The single crystal X–ray diffraction experiment was also performed with the orange diamond–shaped single crystal to determine the structure of the probe (Appendix A). The QNH crystallizes in tetragonal space group *I 41/a*.

### 2.2. Crystal Structure Description

The probe QNH has been crystallized by diffusion of diethyl ether into the chloroform solution of QNH. The single crystal found was orange in colour and a diamond shape. QNH crystallizes in tetragonal space group *I 41/a*. From the crystal structure we found that the quinoxaline moiety and naphthalene moiety exist in the same plane and a chloroform molecule found as solvent. The two-imine bond length C12–N2 is 1.321 Å and C11–N1 is 1.307Å, which we found to be similar to that reported in the literature. The bond angle C12–N2–N1 is 113.40° and C11–N1–N2 is 112.39°, which indicates the *sp^2^* hybridized the N1 and N2 atoms. All carbon and nitrogen atoms are *sp^2^* hybridized and only one oxygen is *sp^3^* hybridized. From the supramolecular 2D structure, the moiety chain has been progressed by intermolecular H–bonding between O1···H15–C15 (Appendix A, Appendix A) and a short interaction between two probe units, by quinoxaline and naphthaldehyde moiety (Appendix A, Appendix A). The extended supramolecular 2D packing diagram has been shown in Appendix A. The QNH + Cu^2+^ complex crystallizes in triclinic space group P −1 and has two distinct units (labelled as A and B) (Appendix A). In both units Cu^2+^ binds in a penta-coordinate way with one oxo oxygen (O1) of naphthalene moiety, one nitrogen (N1) of imine bond, one nitrogen (N3) of quinoxaline moiety, and two DMF solvent moiety.

### 2.3. Naked Eye Sensing

After the characterization of the probe, the initial testing of the probe has been performed in the presence of different metal ions (M^n+^ = Na^+^, K^+^, Ca^2+^, Mg^2+^, Cr^3+^, Cd^2+^, Hg^2+^, Pb^2+^, Mn^2+^, Fe^3+^, Co^2+^, Ni^2+^, Cu^2+^, Zn^2+^ and Al^3+^) available in their chloride and perchlorate salts in HEPES−buffer/acetonitrile (3/7, *v*/*v*) medium at room temperature. The probe shows a distinct colour change from colourless to purple, visible to bare eyes, only in the presence of the Cu^2+^ ion in HEPES−buffer/acetonitrile (3/7, *v*/*v*) medium (Figure 1). Depending on this response, the further spectral studies of the probe in the presence of the Cu^2+^ ion has been performed further in HEPES−buffer/acetonitrile (3/7, *v*/*v*) medium at room temperature.

### 2.4. Photophysical Studies of QNH towards Cu^2+^

The UV–Visible spectra of the probe QNH (10^–5^ M) shows peak intensity at 477 nm and 425 nm. The UV–Visible spectra of the probe in the presence of various metal ions (Na^+^, K^+^, Ca^2+^, Mg^2+^, Hg^2+^, Ni^2+^, Fe^3+^, Cu^2+^, Co^2+^, Cd^2+^, Zn^2+^, Mn^2+^, Pb^2+^, Al^3+^, and Cr^3+^) are recorded in HEPES−buffer/acetonitrile (3/7, *v*/*v*) medium. The probe QNH changes its colour from colourless to purple (Figure 1) in the presence of the Cu^2+^ ion, and in spectral behaviour a new peak at 552 nm had appeared (Appendix A). The probe QNH did not show any selectivity towards anions in the UV–Visible spectrum.

To properly investigate the sensing behaviour of QNH in the presence of Cu^2+^, the titration experiment was performed. Upon gradual addition of Cu^2+^, the 477 nm and 425 nm characteristic peaks of the probe are gradually decreased, and a new peak is generated at 552 nm with an isosbestic point at 484 nm. The peak at 552 nm reached its maxima upon addition of 5 equivalent of Cu^2+^ solution (Figure 2). The isosbestic point indicates there is equilibrium between the probe and the probe in the presence of Cu^2+^. It also indicates that only one species has been generated from the probe upon binding with Cu^2+^. The shifting of peak maxima of the probe from 425 nm to 552 nm upon addition of Cu^2+^ might be explained by intra ligand charge transfer (ILCT) process. The quinoxaline moiety commonly acts as an electron withdrawing segment and the electron-rich naphthalene moiety as a donor, so inside the probe, there may be electron densities in motion. In the Cu^2+^ complex the absorption peak is the result of the ILCT band, due to the fact that the colour change from colourless to purple occurred.

From the UV–Visible titration result, the limit of detection (LOD = 3σ/s, where σ is the standard deviation of the blank solution and s is the slope of the calibration curve) [32] calculated for Cu^2+^ was 4.5 × 10^–7^ M (Appendix A).

To check the selectivity of QNH with Cu^2+^ in the presence of other metal ions, a competitive study had been performed with the probe in the presence of 5 equivalent of other metal ions followed by addition of Cu^2+^ ion. In results, Cu^2+^ shows a selective response among all the cations in HEPES−buffer/acetonitrile (3/7, *v*/*v*) medium (Appendix A). This study indicates the probe can be a selective sensor for Cu^2+^ apart from the other metal ions in semi-aqueous medium.

According to linear Benesi–Hildebrand expression, the measured absorbance [1/(A − A_0_)] at 552 nm varied as a function of 1/[Cu^2+^] in a linear relationship. This indicates the formation of 1:1 stoichiometry between Cu^2+^ and probe QNH. The association constant is found 79.9 × 10^4^ M^–1^ (Appendix A).

The stoichiometric ratio was further investigated by Job′s plot analyses. The molar fraction of the ligand was shown the highest value 0.5 at 552 nm, which also indicates 1:1 stoichiometry between Cu^2+^ and probe QNH (Appendix A). The ESI–MS spectrum of QNH in the presence of Cu shows the 100% peak at 462.06 amu for the species [C_26_H_15_CuN_4_O]^+^ (Appendix A). Both Job’s plot and mass spectrum, the binding stoichiometry of the probe and the probe with Cu^2+^, can be concluded as 1:1 equivalent. The ^1^H NMR spectrum of the probe with Cu^2+^ cannot be recorded due to the paramagnetic behaviour of Cu^2+^.

The optical phenomenon of probe QNH in the presence of Cu^2+^ was checked in different pH levels from 4–10. The effectivity of the probe towards Cu^2+^ was observed from pH range 6–8 and highest in pH 7. This indicates the probe most effectively detects Cu^2+^ in physiological pH (Appendix A).

The absorbance of probe QNH has been checked in the presence of different anions of Cu^2+^ salts and found that QNH effectively shows the same result for all the different forms of Cu^2+^ ion (Appendix A).

The reversibility test of QNH + Cu^2+^ solution was checked in the presence of EDTA in HEPES−buffer/acetonitrile (3/7, *v*/*v*) medium. In the presence of EDTA, the characteristic 552 nm peak of the QNH + Cu^2+^ complex had disappeared, and only peaks of QNH with 477 nm and 425 nm appeared, indicating reversible coordination between QNH and Cu^2+^ (Appendix A). This reversibility experiment has been studied further in several cycles with a sequential addition of QNH + Cu^2+^ solution and EDTA. From the result, it can be concluded that QNH is not a chemodosimeter in the presence of Cu^2+^ and the probe can be recovered even after the experiment.

### 2.5. Theoretical Study and the Elucidation of a Proposed Model

DFT and time-dependent DFT (TD−DFT) calculations were performed to assess the nature of the excited states that govern the observed changes in the absorption spectra. In the implemented computational protocol, the structure of the QNH probe was fully optimized in the absence and the presence of Cu^2+^. To be consistent with the experimental evidence, the latter chemical model accounts for the probe in its deprotonated form. The theoretical absorption wavelength for the QNH (471 nm) and the QNH−Cu^2+^ counterpart (525 nm) agrees with the experimental profile. It is known that the interaction of the metal ions with sensors might yield metal-to-ligand charge transfer (MLCT), ligand-to-metal charge transfer (LMCT), and intra-ligand charge transfer (ILCT) mechanisms. The analysis of the vertical transitions reveals that in both systems, these two characteristic peaks correspond to the HOMO → LUMO transition, which is associated with an ILCT transition (see Figure 3).

### 2.6. Absorption Spectroscopic Studies of QNH–Cu^2+^ Complex towards Histidine

Amino acids are the main building block of proteins. Several works have reported about the affinity of Cu^2+^ with different amino acids [25,33]. Therefore, we have studied the effect of amino acid towards our Cu^2+^ complex. The absorption study of the complex has been carried out in the presence of different amino acids (Gly, Ala, Phe, Val, Ser, Cys, Leu, Ile, Pro, Lys, His, Trp) in HEPES−buffer/acetonitrile (3/7, *v*/*v*) medium (Figure 4). In the presence of histidine, the absorption peak of Cu^2+^ complex at 552 nm has disappeared and the absorption spectra has a similarity with the QNH probe itself. The colour changes from purple to colourless, visible to the naked eye.

Furthermore, we have performed the UV–Visible titration experiment of QNH–Cu^2+^ complex in the presence of histidine to investigate the binding properties. Upon gradual addition of histidine, the complex peak at 552 nm gradually decreases. The isosbestic point at 488 nm indicates the presence of an equilibrium between QNH–Cu^2+^ complex and histidine (Appendix A). The detection limit (LOD = 3σ/s, where σ is the standard deviation of the blank solution and s is the slope of the calibration curve) for histidine has been found 1.35 × 10^−6^ M (Appendix A). From the UV–Visible experiments, we noticed the similarity between the probe and QNH–Cu^2+^ complex with histidine. Due to high affinity between Cu^2+^ and histidine, the Cu^2+^ binds readily with histidine residue, leaving the probe molecule free. The schematic representation has been depicted in Figure 1.

### 2.7. Application

To check the feasibility of the probe in other solvent mediums, we also checked our probe in solid and gel mediums. The gel has been prepared as the method described in literature using Poloxamer–407 [34,35]. Furthermore, in both the phases, the probe QNH shows a colour change from colourless to purple in the presence of Cu^2+^. The result, visible to the naked eye, has been shown in Appendix A.

### 2.8. Proposed Sensing Mechanism of Cu^2+^ by QNH

Quinoxaline moiety has been known for its good colorimetric sensing behaviour. From the experimental findings, there are 1:1 interactions present between the probe QNH and Cu^2+^, and the colour change appears from colourless to purple. From the molecular structure of the complex, we found that the deprotonation of −OH group of QNH happened and the Cu^2+^ ion bind in a penta-coordinate way by two nitrogen, one oxygen and two solvent molecule in 1:1 ratio. Upadhyay et al. also reported a quinoxaline−salicylaldehyde based probe for Cu^2+^ in ethanol: H_2_O medium [28]. Here, due to the presence of naphthalene moiety, the electron density of the overall molecule resides from naphthalene to quinoxaline. Therefore, the presence of naphthalene moiety increases the donor acceptor extended conjugation between the probe, which affects binding with the Cu^2+^ ion; a strong intra ligand charge transfer (ILCT) band appears at 552 nm (Figure 5) in agreement with the theoretical value predicted by TD-DFT (525 nm).

## 3. Conclusions

Our contribution touches upon a problem of interest in analytical chemistry, e.g., the design of selective and sensitive determination methods of copper (II) [36,37,38,39,40,41,42]. We have performed the follow-up work of quinoxaline–naphthaldehyde-based probe QNH and its selective detection of Cu^2+^ in HEPES−buffer/acetonitrile (3/7, *v*/*v*) medium. The probe has been characterized by IR, NMR, and ESI–MS spectroscopy. This work is the elaborate description of the affinity of QNH towards Cu^2+^. The spectral and theoretical findings suggest the binding of Cu^2+^ enhance the ILCT mechanism pathway. The reversibility of probe QNH was also studied as it can be easily recovered and reused. The affinity towards histidine is also discussed here. The crystal structure of probe QNH in CHCl_3_ and QNH + Cu^2+^ complex has been produced and characterized by single crystal X-ray diffraction method. The tune of the absorption band by the interaction of the probe with Cu^2+^ ion is reproduced through TD–DFT, which further supports the proposed sensing mechanism. That agreement confirms DFT as a valuable tool in the design of more efficient probes in the future.

## 4. Experimental Section

### 4.1. Materials and Physical Methods

Ninhydrin, *o*–phenylenediamine, 2−hydroxy naphthaldehyde, and hydrazine hydrate were purchased from Sigma–Aldrich, Kolkata, India and used as received. The salts of cations were also purchased from Sigma–Aldrich, Kolkata, India. Solvents for the syntheses were purchased from commercial sources and used as received. ^1^H NMR spectra were recorded in CDCl_3_ with TMS as internal standard on a Bruker, AV300 Supercon Digital NMR system with dual probe. The FT–IR spectra were recorded from KBr pellets in the range of 400−4000 cm^–1^ on a Perkin−Elmer Spectrum 100 spectrometer (Perkin-Elmer, Shelton, CT, USA). Elemental analyses for C, H, and N were performed on Perkin–Elmer 2400 II analyzers (Perkin-Elmer, Waltham, MA, USA). The ESI-MS experiments were performed on a Waters Xevo G2-S QTOF mass spectrometer (Waters, Milford, CT, USA). The crystallographic data was recorded on a Bruker Nonius Apex II CCD diffractometer (Bruker, Karlsruhe, Germany). The absorption spectral studies were performed on a Hitachi UV–Visible U–3501 spectrophotometer (Hitachi, Fukuoka, Japan).

### 4.2. Single-Crystal X-Ray Crystallography

A high-quality single crystal of QNH was chosen and mounted on a Bruker Nonius Apex II CCD diffractometer equipped with a graphite monochromator and Mo-K_α_ (λ = 0.71073Å) radiation. The diffraction data were collected with exposure time of 5 s/frame and at 296 K. The structure was solved by direct methods and refined by full–matrix, least–squares techniques based on *F^2^*, using SHELXL−2016/6software package [43]. A multi–scan empirical absorption correction was applied using the SADABS program [44]. All non–hydrogen atoms were refined in the anisotropic approximation against *F^2^* of all reflections and determined from the difference Fourier maps. All the hydrogen atoms are isotropically refined. The crystallographic figures have been generated using Mercury 3.9 and Diamond 3.0 codes [45]. The structure refinement parameters and crystallographic data of ligand QNH and its copper complex are listed in Appendix A. CCDC 2156635 and 2156636 contain the crystallographic data in CIF format. This data can be obtained free of charge from the Cambridge Crystallographic Data Centre via www.ccdc.cam.ac.uk/datarequest/cif.

### 4.3. Computational Details

*Ab initio* calculations have been previously used for both copper (I) and copper (II)-based compounds [46,47,48]. Herein, we adapt a similar protocol by using the Gaussian 16 suite of programs [49]. The geometries of both the QNH and QNH−Cu^2+^ model systems were fully optimized within the DFT framework. These model systems used crystal structures as the starting material. The M06 functional [50] was implemented in combination with the def2−SVP basis set to describe all the atoms except the Cu^2^ centre, which was treated with def2−ECP [51]. That latter basis set replaces the core electrons with the effective core potentials to account for scalar relativistic effects without increasing the computational cost [52]. A vibrational analysis is subsequently conducted to ensure that all optimization processes correctly lead to stable minima (no imaginary frequencies). The optimized structures are eventually used to simulate vertical transition energies and compute the frontier molecular orbitals. Solvent effects were included to account for the effect of the acetonitrile environment. An implicit approach was used, the well-known Polarizable Continuum Model (PCM) [53]. We stress that a systematic assessment of the TD-DFT performance is a prerequisite for mimicking chemosensor, as model systems need to be designed with a heterogenic series of metallic centres. However, the selection of that specific level of theory is guided by an available benchmarking [54], and consequently M06 is accurate enough for our purposes.

### 4.4. Synthesis of the Ligand (QNH)

Indeno[1,2−b]quinoxalin−11−one (compound **1**) and 2−hydroxy naphthaldehyde hydrazide (compound **2**) were prepared from the literature procedure [28,55]. In methanolic solution (10 mL) of indeno[1,2–b]quinoxalin–11–one (116 mg, 0.5 mmol), 2−hydroxy naphthaldehyde hydrazide (93 mg, 0.5 mmol) in methanol was added dropwise and refluxed for 2 h (Appendix A). The orange-coloured precipitate was filtered through suction filtration and air dried, yield = 73% (146 mg, 0.36 mmol). Anal. calc. for C_26_H_16_N_4_O: C, 77.99; H, 4.03; N, 13.99; Found: C,77.2; H,4.02; N,13.1; ^1^H NMR (300 MHz, CDCl_3,_ 290 K, TMS) δ_H_ (ppm): 14.14 (s, −OH), 9.86 (s, 1H), 8.27−7.78 (m, 8H), 7.64−7.33 (m, 6H); FT–IR (KBr, cm^−1^): 3066, 2945, 1903, 1623, 1576, 1528, 1460, 1392, 1336, 1288, 1233, 1185, 1113, 1073, 1025, 958, 870, 818, 734, 535; ESI−MS (*m/z*), ion: Calculated: 400.13 amu, Found: 401.14 amu [QNH + H^+^].

### 4.5. Synthesis of the Cu^2+^ Complex (QNH)

0.01 mmol (4 mg) QNH was dissolved in 5 mL 1:1 DMF:methanol. Then, 0.01 mmol (3.7 mg) copper perchlorate hexahydrate was added. The colour changes from light yellow to purple instantly. The solution was then refluxed for 4 h. Afterwards, the reaction mixture was cooled to room temperature and was layered with diethyl ether as solvent. After 4 weeks, rectangular cuboid-shaped black bronze-coloured crystal came on the side walls of the layering tube. ESI−MS (*m/z*) Calculated: 462.06 Found: 462.061 [C_26_H_15_CuN_4_O]^+^ (Appendix A). FT–IR (KBr, cm^−1^): 3054, 1606, 1522, 1451, 1423, 1346, 1198, 1114, 1072, 1030, 763, 619. (Appendix A)

### 4.6. Sample Preparation for Spectroscopic Studies

All the stock solutions were prepared using spectroscopy graded acetonitrile and working solutions were prepared using spectroscopy graded acetonitrile and HEPES buffer. The spectral studies of the probe QNH was carried out in 3:7 HEPES buffer (pH = 7.4, 25 mM): acetonitrile medium. All experiments were carried out at room temperature of 298 K. The stock solutions of the cations were prepared by the perchlorate salts. In the UV–Visible experiment, a stock solution of 10^−3^ (M) QNH was filled in a quartz optical cell of 1.0 cm optical path length to achieve a final concentration of the solution of QNH (10^−5^ (M) in 2000 µL. After 1 min, each spectral data was recorded with the addition of cation solution by using a micropipette.

## Data Availability

The data presented in this study are available upon request from the corresponding author.

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
