# Peer review of "A Quinoxaline−Naphthaldehyde Conjugate for Colorimetric Determination of Copper Ion"

_molecules, 2022, doi:10.3390/molecules27092908_

Round 1

Reviewer 1 Report

Goswami, Cerón-Carrasco and coworkers report a rather comprehensive study combining analytical and computational chemistry. The work revolves around the synthesis of a new organic QNH ligand which shows a high affinity for binding with copper. The authors, therefore, describe the free complex using ESI-MS, NMR, UV-VIS, elemental analysis and XRD and its copper adduct. Simultaneously they do optimisation and calculation of the excitation spectrum with DFT, TD-DFT respectively. The interpretations are is a standard alignment with the observations.

Overall, the study may fulfil the rigour or the interest of the analytical and computational chemists. However, as molecules are rather open to all sorts of chemists, coordination chemists may be critical about certain aspects of the work. In this regard, I want to point out clearly what the authors should address prior to considering the article for publication:

  1. The XRD study shows that the complexed copper(II) is 5-coordinate. This is within the expectations. However, when the DFT and TD-DFT modelling is done, the structures taken are of 3-coordinate copper. In the solid state, the two oxo-providing ligands are bulky and are parts of other molecules. Modelling them would be difficult, and I fully understand that the authors omitted that part so that the 3-coordinate structure remains robust and thus it converges more optimally and it does not produce imaginary frequencies. However, that may not be a reasonable model. In that regard, my recommendation is to model the 5-coordinate complex by using proxy ligands to what has been observed experimentally. This may be a monodentate ligand (e.g. methoxy) or bidentate ligand (e.g. oxalate based).
  2. Another point from the computations is about selective binding. The question is, what is the most fundamental reason why copper has a much stronger affinity to bind in comparison to so many other metal centres?
  3. Figure 1 shows that the QNH ligand is very selective and only copper shows to provide a visible reaction. What is the concentration of the metals in all of those vials? Are they the same?
  4. Scheme 1 is supposed to be a form of a reaction, but the plus signs are missing. Further on, the provided description is not in alignment with what is known on the binding between histidine and copper (see DOI: 10.1016/j.ccr.2004.09.013).
  5. The authors have done crystallographic work. Also mentioned that the crystal structures are available on CCDC (that is numbers 2156635, 2156636), but after a careful check they are not.

The work overall seems to be of potential publication interest. However, considering the existing issues, I believe a major correction is needed.

Author Response

Comment: Goswami, Cerón-Carrasco and coworkers report a rather comprehensive study combining analytical and computational chemistry. The work revolves around the synthesis of a new organic QNH ligand which shows a high affinity for binding with copper. The authors, therefore, describe the free complex using ESI-MS, NMR, UV-VIS, elemental analysis and XRD and its copper adduct. Simultaneously they do optimisation and calculation of the excitation spectrum with DFT, TD-DFT respectively. The interpretations are is a standard alignment with the observations.

Overall, the study may fulfil the rigour or the interest of the analytical and computational chemists. However, as molecules are rather open to all sorts of chemists, coordination chemists may be critical about certain aspects of the work. In this regard, I want to point out clearly what the authors should address prior to considering the article for publication.

Reply: We thank Reviewer 1 for her/his positive comments.

Comment: The XRD study shows that the complexed copper(II) is 5-coordinate. This is within the expectations. However, when the DFT and TD-DFT modelling is done, the structures taken are of 3-coordinate copper. In the solid state, the two oxo-providing ligands are bulky and are parts of other molecules. Modelling them would be difficult, and I fully understand that the authors omitted that part so that the 3-coordinate structure remains robust and thus it converges more optimally and it does not produce imaginary frequencies. However, that may not be a reasonable model. In that regard, my recommendation is to model the 5-coordinate complex by using proxy ligands to what has been observed experimentally. This may be a monodentate ligand (e.g. methoxy) or bidentate ligand (e.g. oxalate based).

Reply: As correctly noted by the referee, copper(II) is penta-coordinate entity. Consequently, our calculations for the QNH+Cu2+ system has been redone by using to mimic that coordination sphere. Figures has been rendered again to show the associated optimized structure. The optical profile computed with the 5-coordinate model systems is also included in the main text. Computational details specified now that X-ray structures are used as starting geometries for optimizations. All these additional calculations are consistent with conclusions reached in the original version of our manuscript.

Comment: Another point from the computations is about selective binding. The question is, what is the most fundamental reason why copper has a much stronger affinity to bind in comparison to so many other metal centres?

Reply: We fully agree with Reviewer 1. Theory might help to rationalize probe-metal interactions with a focus on designing novel molecules. However, this is not a trivial task, which is particularly true if several metallic centers need to be modeled (and compared). We are positive that a systematic benchmarking of DFT methods (or higher levels of theory) by using several probes and metals will be welcomed by Molecules readers. However, such large benchmarking is out of the scope of the present manuscript. Herein, DFT is specifically focus on the assessment of the QNH probe to confirmation of the proposed mechanism.

Revised manuscript includes now a wider view of the available literature, including previous DFT works with copper and a more critical assessment about required next theoretical steps.

Comment: Figure 1 shows that the QNH ligand is very selective and only copper shows to provide a visible reaction. What is the concentration of the metals in all of those vials? Are they the same?

Reply: The concentration of the metals in all of the vials are 10-4 M. This is now specified in the caption of Figure 1.

Comment: Scheme 1 is supposed to be a form of a reaction, but the plus signs are missing. Further on, the provided description is not in alignment with what is known on the binding between histidine and copper (see DOI: 10.1016/j.ccr.2004.09.013).

Reply: We apologize for the inconvenience. The plus signs have been added to Scheme 1. Unfortunately, we do not have the crystal structure for the copper-histidine moiety and several binding modes can be possible as described in the literature suggested by reviewer(DOI: 10.1016/j.ccr.2004.09.013). Scheme 1 labels now the product as “His-Cu2+”.

Comment: The authors have done crystallographic work. Also mentioned that the crystal structures are available on CCDC (that is numbers 2156635, 2156636), but after a careful check they are not.

Reply: We now uploaded the cif files and checkcif documents as supplementary files.

Reviewer 2 Report

The work is technically well done and cover an important branch of analytical chemistry (a selective and very sensitive determination of copper(II)). However, while reading this manuscript I have found one very important question.

The main question to this material: the authors need to explain in more details what is the advantage of the method they have developed. A tremendous amount of papers (including the last 3-4 years) is devoted to the selective colorimetric determination of copper ions by complexation with different organic molecules. Moreover, in many articles, the limit of detection (LOD) of copper is much lower than in the present work (up to 10 times).  Some of the early works (but by no means all of them!) are summarized in Table S1. However, the present article does not compare the original results with earlier papers. Also, the authors have lost some other important publications in the Introduction and Discussion section. These papers include rather different but very effective systems for the selective colorimetric determination of copper ions. See, for example, the following references:

- Meijun Li, Xiaoxuan Huang, Haili Yu, Materials Science and Engineering: C, 2019, Vol. 101, P. 614-618, https://doi.org/10.1016/j.msec.2019.04.022

- Zahra Shojaeifard, Bahram Hemmateenejad, Mojtaba Shamsipur, Raheleh Ahmadi, Journal of Photochemistry and Photobiology A: Chemistry, 2019, Vol. 384, 112030, https://doi.org/10.1016/j.jphotochem.2019.112030

- Xiaogen Chen, Qiujun Lu, Dan Liu, Cuiyan Wu, Meiling Liu, Haitao Li, Youyu Zhang, Shouzhuo Yao, Microchimica Acta, 2018, Vol. 185, 188, https://doi.org/10.1007/s00604-018-2720-y

- Gopi Kalaiyarasan, James Joseph, Analytical and Bioanalytical Chemistry, 2019, Vol. 411, P. 2619–2633, https://doi.org/10.1007/s00216-019-01710-8

- Asadollah Mohammadi, Behzad Khalili, Aida Saberi Haghayegh, Spectrochimica Acta Part A: Molecular and Biomolecular Spectroscopy, 2019, Vol. 222, 117193, https://doi.org/10.1016/j.saa.2019.117193

- Yuanyuan Cao, Yingnan Liu, Fan Li, Shuwen Guo, Yuhang Shui, Hanyue Xue, Li Wang, Microchemical Journal, 2019, Vol. 150, 104176, https://doi.org/10.1016/j.microc.2019.104176

- Leila Khoshmaram, Masoud Saadati, Ali Karimi, Analytical Methods, 2020, Vol. 12, P. 3490-3498, https://doi.org/10.1039/D0AY00706D

Moreover, the Cu(II) chemosensor based on quinoxaline and naphthaldehyde based Schiff base (QNH) has already been proposed by another group (the ref. [14b]). The authors need to explain the importance of their system. Without more extended discussion and comparison with previous results, this material looks fragmented and does not bring new inside this theme.

Some other technical moments:

1. The ESI−MS spectrum of QNH (Fig. S3) contains a number of peaks with m/z much higher than the value for a molecular ion. It should be explained! At present, it looks like the sample is not pure. The difference of the found carbon and nitrogen content in the elemental analysis (page 8, lines 288-289) from the calculated values may also say about some impurities in the sample.

2. Authors say that "Elemental analyses for C, H and N were per-249 formed on a Perkin–Elmer 2400 II analyzers." (page 7, lines 249-250). However, on page 8, the lines 288-289, the elemental analysis is given as the following: "Anal. calc. for C26H16N4O : C, 77.99; H, 4.03; N, 13.99; O, 4; Found : C,77.2; H,4.02; N,13.1; O,4" The oxygen content was also determined? If yes, then how?

3. The X-ray experiments seem to be of very low quality. For example, R1 and wR2 values [I > 2σ(I)] are 0.1034 and 0.3079 for the crystallographic data for QNH!, and 0.1842, 0.4651 for QNH + Cu(II)! GOOF is more than 2 for the second experiment. From this point of view, the discussion about bond lengths etc. (page 2) looks like speculation. The accuracy of Cg∙∙∙Cg distances like 3.7606(10)Å does not matter. Please check! For a more detailed evaluation of the quality of X-ray experiments,  the authors should provide cif-files and check-cif results for future consideration.

4. The Reference section should be made in the accordance with the rules of Molecules.

Author Response

Comment: The work is technically well done and cover an important branch of analytical chemistry (a selective and very sensitive determination of copper(II)). However, while reading this manuscript I have found one very important question.

The main question to this material: the authors need to explain in more details what is the advantage of the method they have developed. A tremendous amount of papers (including the last 3-4 years) is devoted to the selective colorimetric determination of copper ions by complexation with different organic molecules. Moreover, in many articles, the limit of detection (LOD) of copper is much lower than in the present work (up to 10 times).  Some of the early works (but by no means all of them!) are summarized in Table S1. However, the present article does not compare the original results with earlier papers. Also, the authors have lost some other important publications in the Introduction and Discussion section. These papers include rather different but very effective systems for the selective colorimetric determination of copper ions. See, for example, the following references:

- Meijun Li, Xiaoxuan Huang, Haili Yu, Materials Science and Engineering: C, 2019, Vol. 101, P. 614-618, https://doi.org/10.1016/j.msec.2019.04.022

- Zahra Shojaeifard, Bahram Hemmateenejad, Mojtaba Shamsipur, Raheleh Ahmadi, Journal of Photochemistry and Photobiology A: Chemistry, 2019, Vol. 384, 112030, https://doi.org/10.1016/j.jphotochem.2019.112030

- Xiaogen Chen, Qiujun Lu, Dan Liu, Cuiyan Wu, Meiling Liu, Haitao Li, Youyu Zhang, Shouzhuo Yao, Microchimica Acta, 2018, Vol. 185, 188, https://doi.org/10.1007/s00604-018-2720-y

- Gopi Kalaiyarasan, James Joseph, Analytical and Bioanalytical Chemistry, 2019, Vol. 411, P. 2619–2633, https://doi.org/10.1007/s00216-019-01710-8

- Asadollah Mohammadi, Behzad Khalili, Aida Saberi Haghayegh, Spectrochimica Acta Part A: Molecular and Biomolecular Spectroscopy, 2019, Vol. 222, 117193, https://doi.org/10.1016/j.saa.2019.117193

- Yuanyuan Cao, Yingnan Liu, Fan Li, Shuwen Guo, Yuhang Shui, Hanyue Xue, Li Wang, Microchemical Journal, 2019, Vol. 150, 104176, https://doi.org/10.1016/j.microc.2019.104176

- Leila Khoshmaram, Masoud Saadati, Ali Karimi, Analytical Methods, 2020, Vol. 12, P. 3490-3498, https://doi.org/10.1039/D0AY00706D

Reply: We first thanks Reviewer 2 for his/her positive comments. Colorimetric determination of metal ions is an accurate but fast detection method if correctly designed. In that framework, and as correctly raised by Reviewer 2, the quinoxaline moiety has been known as an excellent colorimetric signaling unit. The probe in naked eye, can detect Cu2+ in HEPES−buffer/acetonitrile (3/7, v/v) medium very rapidly and the distinct color change was colorless to purple. In case of detection limit (even if not the lowest in the literature), it is quite lower than WHO guidelines, and allow for a quick determination.

All suggested references suggested by the reviewer are now included in the revised manuscript.

Comment: Moreover, the Cu(II) chemosensor based on quinoxaline and naphthaldehyde based Schiff base (QNH) has already been proposed by another group (the ref. [14b]). The authors need to explain the importance of their system. Without more extended discussion and comparison with previous results, this material looks fragmented and does not bring new inside this theme.

Reply: Das and co-workers have reported the ligand system and its study in methanol/aqueous HEPES (9:1, v/v) medium. Here we have done the study in HEPESbuffer/acetonitrile (3/7, v/v) and found different results. Though the ligand system was earlier reported, we provided a more complete picture. The crystal structure of QNH-Cu2+confirmed the binding mode of the probe with Cu2+. DFT and TD-DFT studies were also performed. All these accumulated new findings were not available in that earlier work.

Comments: The ESI−MS spectrum of QNH (Fig. S3) contains a number of peaks with m/z much higher than the value for a molecular ion. It should be explained! At present, it looks like the sample is not pure. The difference of the found carbon and nitrogen content in the elemental analysis (page 8, lines 288-289) from the calculated values may also say about some impurities in the sample.

Reply: The ESI−MS spectrum of the sample was recorded in very dilute acetonitrile medium. We found 100% peak result for our ligand. As correctly noted by the referee, other minor peaks are present that might be associated to a small amount of other samples. The elemental analysis had been done at the preliminary stage with the crude sample. That minor impurity does not affect to our results, though we do agree that should be commented in the supplementary information, which has been modified accordingly. 

Comments:  Authors say that "Elemental analyses for C, H and N were per-249 formed on a Perkin–Elmer 2400 II analyzers." (page 7, lines 249-250). However, on page 8, the lines 288-289, the elemental analysis is given as the following: "Anal. calc. for C26H16N4O : C, 77.99; H, 4.03; N, 13.99; O, 4; Found : C,77.2; H,4.02; N,13.1; O,4" The oxygen content was also determined? If yes, then how?

Reply: We apologize for the inconvenience. This is a typing error that has been fixed in the revised version.

Comments:  The X-ray experiments seem to be of very low quality. For example, R1 and wR2 values [I > 2σ(I)] are 0.1034 and 0.3079 for the crystallographic data for QNH!, and 0.1842, 0.4651 for QNH + Cu(II)! GOOF is more than 2 for the second experiment. From this point of view, the discussion about bond lengths etc. (page 2) looks like speculation. The accuracy of Cg∙∙∙Cg distances like 3.7606(10)Å does not matter. Please check! For a more detailed evaluation of the quality of X-ray experiments,  the authors should provide cif-files and check-cif results for future consideration.

Reply: We agree with the reviewer for the comment on data quality for XRD experiment. Unfortunately, all tries to enhance X-ray quality crystal fails. The high R1 and wR2 values are caused by the twining, where twins did not overlap exactly. For uncertainty of the atomic position of atoms in XRD experiment we have removed the discussion regarding the metric parameter of QNH + Cu(II) complex from the manuscript. CIF and checkcif of the complexes have been provided as supplementary material of the paper.

Comments:  The Reference section should be made in the accordance with the rules of Molecules.

Reply: The reference section has been rearranged according to Molecules template.

Reviewer 3 Report

This manuscript could be accepted for publication in Molecules. The novelty of presented research and it's importance for the scientific community focused on analytical chemistry are high. This work facilitates detection of bivalent copper ion by a simple Schiff base probe based on quinoxaline−naphthaldehyde framework. The detailed study in absorption spectroscopy and theoretical aspects and crystal study of the probe and probe−copper complex has been discussed. The detection limit of the probe in presence of Cu2+ is 0.45 μM in HEPES−buffer/acetonitrile (3/7, v/v) medium for absorption study. The reversibility of the probe−copper complex has been investigated by EDTA. The selective visual detection of copper has been established also in gel form. The introduction provide sufficient background. The research methodology is adequate and modern. The results are clearly presented. The amount of data is large. The conclusions supported by the data. The manuscript good illustrated and interesting to read. English language and style are fine, and may be very minor polishing from native speaker is recommended. I have also couple of minor suggestions. Firstly, I suggests to cite some relevant papers in Introduction and/or Computational Details sections to show that other researchers also used similar methodology of DFT and TD-DFT calculations for similar chemical compounds (e.g. Inorg. Chem. 2021, V. 60. P. 1149.; Inorg. Chim. Acta 2015, V. 434. P. 31.; Dalton Trans., 2012, V. 41, P. 14157.;). Secondly, it would be useful for potential readers if authors will add Cartesian atomic coordinates for all model structures in Supplementary Materials. Finally, checkcif files for crystallographic data should be provided and not contain alerts level A and B (or these alerts should be commented and justified).
Overall, this manuscript could be accepted for publication after minor revisions.

Author Response

Comment: This manuscript could be accepted for publication in Molecules. The novelty of presented research and it's importance for the scientific community focused on analytical chemistry are high. This work facilitates detection of bivalent copper ion by a simple Schiff base probe based on quinoxaline−naphthaldehyde framework. The detailed study in absorption spectroscopy and theoretical aspects and crystal study of the probe and probe−copper complex has been discussed. The detection limit of the probe in presence of Cu2+ is 0.45 μM in HEPES−buffer/acetonitrile (3/7, v/v) medium for absorption study. The reversibility of the probe−copper complex has been investigated by EDTA. The selective visual detection of copper has been established also in gel form. The introduction provide sufficient background. The research methodology is adequate and modern. The results are clearly presented. The amount of data is large. The conclusions supported by the data. The manuscript good illustrated and interesting to read. English language and style are fine, and may be very minor polishing from native speaker is recommended.

I have also couple of minor suggestions. Firstly, I suggests to cite some relevant papers in Introduction and/or Computational Details sections to show that other researchers also used similar methodology of DFT and TD-DFT calculations for similar chemical compounds (e.g. Inorg. Chem. 2021, V. 60. P. 1149.; Inorg. Chim. Acta 2015, V. 434. P. 31.; Dalton Trans., 2012, V. 41, P. 14157.;).

Reply: All suggested references are now cited in the revised version of our manuscript

Secondly, it would be useful for potential readers if authors will add Cartesian atomic coordinates for all model structures in Supplementary Materials. Finally, checkcif files for crystallographic data should be provided and not contain alerts level A and B (or these alerts should be commented and justified).

Reply:  We have now uploaded the cif files and checkcif documents as supplementary files for your reference. Cartesian atomic coordinates are added to the Supplementary Materials.

Round 2

Reviewer 1 Report

The authors positively responded to all of my main concerns.

In my view, the article is in a good shape for publication in Molecules. 

Reviewer 2 Report

The authors have addressed all comments given by reviewer, and revised the manuscript in a proper way. I believe it may be accepted for publication. Good luck to authors!